# Increasing Healthy Food Access for Low-Income Communities: Protocol of the Healthy Community Stores Case Study Project

**DOI:** 10.3390/ijerph19020690

**Published:** 2022-01-08

**Authors:** Joel Gittelsohn, Christina M. Kasprzak, Alex B. Hill, Samantha M. Sundermeir, Melissa N. Laska, Rachael D. Dombrowski, Julia DeAngelo, Angela Odoms-Young, Lucia A. Leone

**Affiliations:** 1Department of International Health, Johns Hopkins University Bloomberg School of Public Health, Baltimore, MD 21205, USA; srex2@jh.edu; 2Department of Community Health and Health Behavior, University at Buffalo, Buffalo, NY 14260, USA; cmk27@buffalo.edu (C.M.K.); lucialeo@buffalo.edu (L.A.L.); 3Community Health Interventions Lab, University at Buffalo, Buffalo, NY 14260, USA; 4Urban Studies and Planning and Detroit Food Map Initiative, Wayne State University, Detroit, MI 48202, USA; alexbhill@wayne.edu; 5Division of Epidemiology and Community Health, University of Minnesota, Minneapolis, MN 55455, USA; mnlaska@umn.edu; 6Division of Kinesiology, Health and Sport Studies, College of Education, Wayne State University, Detroit, MI 48202, USA; rachael.dombrowski@wayne.edu; 7Departments of Health Policy Management & Nutrition, Harvard T.H. Chan School of Public Health, Harvard University, Boston, MA 02115, USA; jdeangelo@hsph.harvard.edu; 8Division of Nutritional Sciences, College of Human Ecology, Cornell University, Ithaca, NY 14850, USA; odoms-young@cornell.edu

**Keywords:** food access, healthy food retail, case study approach, urban, qualitative, low-income

## Abstract

Improving healthy food access in low-income communities continues to be a public health challenge. One strategy for improving healthy food access has been to introduce community food stores, with the mission of increasing healthy food access; however, no study has explored the experiences of different initiatives and models in opening and sustaining healthy food stores. This study used a case study approach to understand the experiences of healthy food stores in low-income communities. The purpose of this paper is to describe the methodology used and protocol followed. A case study approach was used to describe seven healthy food stores across urban settings in the U.S. Each site individually coded their cases, and meetings were held to discuss emerging and cross-cutting themes. A cross-case analysis approach was used to produce a series of papers detailing the results of each theme. Most case studies were on for-profit, full-service grocery stores, with store sizes ranging from 900 to 65,000 square feet. Healthy Food Availability scores across sites ranged from 11.6 (low) to 26.5 (high). The papers resulting from this study will detail the key findings of the case studies and will focus on the challenges, strategies, and experiences of retail food stores attempting to improve healthy food access for disadvantaged communities. The work presented in this special issue will help to advance research in the area of community food stores, and the recommendations can be used by aspiring, new, and current community food store owners.

## 1. Introduction

Improving healthy food access in low-income communities remains an ongoing challenge throughout the world [1,2]. Low-income urban and rural communities frequently have low access to healthy food retail options and an overabundance of unhealthy options [2,3]. Various approaches have been used to address this inequity in food access in the United States (U.S.). At the federal level, policy revisions to national food assistance programs have broadened the inclusion of healthy foods through the addition of healthier foods within the Women, Infants, Children (WIC) program and by providing healthy food incentives for produce (e.g., Supplemental Nutrition Assistance Program (SNAP) matching programs) [4,5,6]. The federal Healthy Food Financing Initiative has also provided funding to open or expand new food retail in underserved communities. At the city or regional level, policies such as sugar-sweetened beverage (SSB) and junk food taxes [7,8,9], supermarket financing initiatives [10,11], and staple foods ordinances have been tested [12,13]. Within neighborhoods, considerable attention has been paid to developing, implementing, and evaluating interventions in small retail food stores voluntarily participating in these programs [14].

While there has been a push both nationally and locally to entice new supermarkets to open in low food access areas, research on the introduction of new supermarkets in underserved communities has not demonstrated positive impacts on the dietary outcomes of lower-income individuals [15]. There are many possible reasons for the failure of these stores to improve people’s diets; research indicates that people do not always shop at the closest grocery store, and that store ownership is more predictive of purchasing than proximity to healthy food. Furthermore, while supermarkets offer a variety of foods, including fresh produce, they also offer many unhealthy items, and food quality may be lower in certain communities. Another possibility is that residents do not feel that they have a say or buy-in when a new store is brought to their community. Many of these stores fail entirely and go out of business. For example, one paper reviewed studies on grocery stores that opened in neighborhoods with low access to healthy food that ultimately failed. Some of the reasons stated for their closure included poor sales, inadequate marketing, and a lack of food retail experience. This review highlighted that the reasons for store closures in this context are understudied and not reported in great detail, especially in urban settings [16].

As an alternative, the introduction and/or support of “healthy food stores” to lower income communities may be a promising strategy for improving healthy food access. By the term “healthy food stores”, we differentiate these stores from traditional grocery and retail food sources in that their primary mission is the provision of a range of affordable healthier foods. Such stores may be for profit or non-profit, but would ultimately be self-sustaining and require limited outside support. More importantly, traditional retail food stores may change to adopt this primary mission of healthier food provision and become healthy food stores.

Despite the interest in such stores, no study has comprehensively explored the experiences of different initiatives and models in such healthy food stores, or has drawn forth common strategies for creating a sustainable healthy food retail environment in lower income communities. In addition, the ongoing COVID-19 pandemic has put unprecedented stress on the U.S. food system, placing a high burden on food distributors and retailers and profoundly affecting consumer access. These pressures are most acutely felt among marginalized low-income populations that commonly lack access to supermarkets, resources for online purchasing, reliable transportation, and financial stability to support bulk purchasing [17,18,19,20,21]. Despite increasing food system disruptions, little research has examined the impact of the pandemic on these healthy food access mission-driven stores [22].

In June 2020, the Robert Wood Johnson Foundation (RWJF) Healthy Eating Research (HER) program funded a study on mission-driven healthy food stores. The goals of the study were: (1) To construct case studies of diverse healthy food-focused retail food stores located in low-income communities throughout the US using a mixed methods approach; (2) To understand the experiences of these stores in the aftermath of the COVID-19 pandemic, including strengths and vulnerabilities; and (3) To conduct a cross-case analysis to understand common strategies for success and the challenges experienced by these stores, eliciting strategies for store survival under usual conditions of food insecurity as well as under the circumstances of the pandemic.

This paper will describe the case study methodology used and provide context for the study itself. We seek to answer the following research questions:What were the origins of the healthy stores case study project? What was the justification for conducting a case study analysis?What was the process used for conducting this research?

## 2. Methods

### 2.1. Origins of the Healthy Stores Case Study Project

Many of the researchers for this study are members of the RWJF HER/NOPREN Healthy Food Retail (HFR) Working Group. The working group is a collaborative effort of the Robert Wood Johnson Foundation’s Healthy Eating Research (HER) program and the Centers for Disease Control and Prevention’s (CDC) Nutrition and Obesity Policy Research and Evaluation Network (NOPREN). The overall goal of the working group is to identify the most effective strategies that would shift consumers away from purchasing and consuming unhealthy, energy-dense foods and beverages and drive them instead toward purchasing and consuming healthier foods and beverages that align with the Dietary Guidelines for Americans (DGA) [23], by designing, conducting, and disseminating research in the food retail setting.

In June 2020, JG developed a proposal in collaboration with the HFR working group to understand the experiences of healthy community food stores in providing healthy food in low-income settings, and to understand what challenges stores faced during the COVID-19 pandemic and how they adapted to these challenges.

### 2.2. Rationale for Taking a Case Study Approach

Case study methodology was selected for this study for several reasons. First, case studies permit depth and the presentation of context by utilizing multiple sources of data to understand and unpack a phenomenon under study [24]. Second, case studies typically employ multiple data collection methods, both qualitative and quantitative, to provide a more complete picture of each case [24]. Due to the diversity of methods, multiple viewpoints are encouraged and can be presented within a case study approach [25,26,27]. Contextual richness is maintained by having findings written up as detailed narrative reports, one for each case (store). Finally, each case study was developed in a different location, by a different study team who know the city and store very well. This built engagement and trust at each site, enhancing the quality of the data collected.

### 2.3. Steps in the Research Process

Our multiple case study used a mixed-methods approach, and followed six steps:

#### 2.3.1. Step 1: Case (Food Store) Recruitment and Selection

A maximum variation sampling approach was used [28]. The HFR WG developed a series of criteria for recruitment to enhance variation, which included: approach for promoting healthy food access, store type (convenience store, grocery store, co-op), business model (for-profit, not for-profit), possession of sufficient capacity to conduct mixed methods research, organizational mission, region (e.g., urban), and demographic served (e.g., low, mixed income). As the primary recruitment strategy, we used the HFR WG listserv, which included over 175 members, to invite the members to nominate themselves and a participating retail food store. Working group members were also encouraged to utilize their outreach channels and share the announcement to facilitate recruitment. Nominations were submitted through a Qualtrics survey, which was administered from mid to late July 2020.

We received nominations for 18 distinct researcher–store partnerships. Funding limited this work to a maximum of 8 case study stores. The main criteria applied in order to narrow the pool of applicants included: stores that served an urban region, stores with a clear mission to improve healthy food access, and research teams with a demonstrated capacity to carry out the research goals. Among the stores that met these criteria, the final choice of stores was based on maximizing geographic (e.g., west coast states) and store type (e.g., co-op, convenience store) heterogeneity. If more than one researcher–store partnership in the same city was nominated, the group that demonstrated a better capacity to participate in research and had a clear mission and history of working together was chosen. This decision was made by the Healthy Food Retail working group. Chosen stores were notified in several waves to ensure the nominators were still interested and able to begin timely participation; one store withdrew from consideration during the decision-making process and was replaced by a similar store type from that city. Due to conflicting organization priorities, one of the eight chosen store–researcher pairs dropped out as project planning commenced.

#### 2.3.2. Step 2: Formation of Case Study Working Group and Selection of Methods for Data Collection

Based on the seven remaining store–researcher partnerships, we formed a new Healthy Stores Case Study (HSCS) working group, chaired by the lead author (JG), with 2–3 researchers associated with each case study. The HSCS WG met every two weeks to develop data collection and analysis protocols. The following methods were selected:In-depth interviews (IDIs) with store owners, managers, and/or staff (n = 3–4/site) regarding origin of the store, community engagement practices, food sourcing, and pricing strategies, both pre-pandemic and during the pandemic, and perceived successes and challenges;Direct observation of the store (including the Nutrition Environment Measure Survey for Stores (NEMS-S) short form) to assess food availability and pricing (3x/store);Perceived unit sales of healthier (e.g., fresh produce) and less healthy items (e.g., sugary beverages and ultra-processed foods), obtained via store manager report (3x/store);IDIs with local stakeholders, designated by the retailer, about ongoing partnerships and the role of the store in the community (n = 3–4/site).

#### 2.3.3. Step 3: Development of IDI Guides

The WG meetings next turned to the development of IDI guides (see Appendix A) and development and refinement of instruments for quantitative data collection [29,30,31,32,33]. The IDI guides were developed through discussion among HSCS members. The most salient overarching categories were chosen (e.g., challenges, adaptations, community engagement), which guided the generation of open-ended questions. The retailer IDI guide (Appendix A) was designed to assess store characteristics, history and mission, store operations pre-2020, operations from 2020 onward, adaptations to the pandemic, overall successes and challenges, perceived impact of the events of 2020 (e.g., pandemic, civil unrest), and community engagement and marketing strategies. The stakeholder IDI guide (Appendix A) was designed to evaluate stakeholders’ relationships with the retailer, perceptions of the store, stakeholders’ involvement in adaptations made by the retailer from 2020 onward, overall success and challenges realized through the partnership, suggestions on how the retailer could improve, and perceived impact of the events of 2020.

IRB approval was obtained when warranted by each research institution.

#### 2.3.4. Step 4: Training of Data Collectors and Data Collection Protocol

Online trainings of data collectors for each site were conducted for the IDIs (led by JG, EL, CK) and for the structured instruments (led by AH) over Zoom. Data collection took place between January and August 2021, with variations by site depending on IRB approval status and logistical issues. During the data collection period, two of the stores (Chicago, Buffalo) dropped from the study (reasons given were the loss of SNAP authorization, too busy) but were replaced with other eligible stores in the community. Each site collected and transcribed its own interviews, and collected and entered quantitative data in a shared database (Qualtrics). Interviewees were provided with a gift card to compensate them for their time and expertise.

#### 2.3.5. Step 5: Preparation of Individual Case Study Narrative Reports at Each Site

Brief descriptions of the seven completed case study stores are presented in Table 1 and Table 2, including store location, type, and size. Most were identified as for-profit, full-service grocery stores. The store sizes ranged from 900 to 65,000 square feet. The average Healthy Food Availability scores across sites ranged from 11.6 to 26.5. A common case study narrative report format for the case studies was developed by the WG and was utilized by each site (Appendix A). The report format specified section titles and headers, and provided a suggested length for each; however, each site had the flexibility to reduce or expand any section based on their data. As data collection was completed, sites compiled their individual case study reports. All seven reports were completed in the period of August–October 2021, and varied in length from 7 to 27 pages. Deidentified versions of these reports are available on request.

#### 2.3.6. Step 6. Cross-Case Analysis of Individual Case Reports and Preparation of Manuscripts

Since the analysis plan is not identical for each project paper, the details of each analysis have been reserved for the manuscripts of these subsequent studies; here, we provide a ‘high level’ overview of the procedures used across the studies.

The analytic approach for cross-case analysis shifted from centralized analyses within each site to cross-site analysis by the full working group. Initially, a coding scheme was developed to be applied to all textual data—with the idea that it would be applied by sites to their own data—and then the results of the query searches were to be reported in an aggregated fashion across sites. A challenge with this approach was the loss of context for each of the cases, a common issue in cross-case analysis. In the end, several sites did use the common coding scheme [34], applying it to their own textual data and identifying key themes and categories, but the coding scheme was not employed for the cross-case analyses, except in one case.

Three of the papers used the multiple case study analytic approach described by Stake [25]. This method facilitates the identification of commonalities and differences across the collective of cases while maintaining a focus on the contextual aspects distinct to each case. Moving through a series of analytic activities (e.g., case familiarization, assessing utility, sorting and merging findings), lead researchers developed assertions about the aggregate of cases to answer their specific research question.

Quantitative data were analyzed based on the standard NEMS-S scoring criteria for the 11 food categories, excluding frozen foods [29]. An amended Healthy Food Access Index was utilized as a primary analysis metric to compare and contrast healthy food accessibility across store sites [35].

As data collection proceeded, the WG developed a series of discussion topics and themes emerging from the data. In meetings, each site presented emergent themes to the group, which were then discussed and compared by researchers from other sites with their own findings. Out of these discussions, a series of paper topics were developed and refined. Lead authors and coauthors were decided by self-nomination, with an effort made to distribute key roles to the seven different sites. The primary analyst for most of the papers was the lead author, supported by 2–3 other coauthors.

### 2.4. Ethical Considerations

Confidentiality of the stores, their employees, and supporting community organizations should be considered in case reporting and manuscript publication. Therefore, the individual case reports in their original format will not be published in order to diminish the risk of breach of confidentiality. Manuscripts published as a result of the cross-case analysis will not include store or employee identifiers. Instead, they will synthesize the lessons learned across stores and discuss barriers and facilitators of store success.

## 3. Results

Seven papers (including this one) are planned subsequent to this study and will focus on the challenges, strategies, and experiences of retail food stores attempting to improve healthy food access for disadvantaged communities. In these papers, we will compare and contrast several different food retail models and experiences.

The first paper following this protocol paper will present the key findings of the study related to food store strategies for success and how to cope with challenges. The second paper will focus on the role of partnership in undergirding the success of these stores, emphasizing community engagement and the role of community-based organizations. That this work took place during the COVID-19 pandemic and has yielded insights into how healthy retailers can be prepared for future emergencies will be the focus of the next paper. The following paper will provide an analysis of the NEMS-S and other quantitative findings, and then compare these with national data. The final paper will summarize the main findings and will comment on needed future directions for this work.

## 4. Discussion

This is the protocol paper for the first case study project on healthy community food stores in urban settings. Seven stores were recruited across diverse urban settings, which allowed for the collection and synthesis of multiple viewpoints and experiences, all of which were used to inform study findings. Subsequently, a toolkit will be created for store owners and researchers to use as a resource for opening a healthy community food store, evaluating its success, and providing models for successfully sustaining a healthy community food store.

Previous food store trials have focused on improving the availability of fresh fruits and vegetables in existing stores [36]. Using mixed methods approaches, these studies identified barriers and facilitators of the stocking and selling of healthier foods in low-income settings, and demonstrated the importance of working with multiple stakeholders [36]. However, other factors that play a role in the success of retail food stores such as business models, funding mechanisms, store operations, and supplier relationships were not captured. The scope of this case study project was broad and aimed to capture these additional elements and expand beyond healthy food access considerations, allowing us to explore in-depth strategies and shortcomings for overall store success. This expanded scope is critical, given that previous work has shown that simply opening a food store in a neighborhood with low food access does not guarantee success or improve community health outcomes [16,37,38].

The study took place during the COVID-19 pandemic, creating challenges for the realization of the study and for the stores as well. The study team was able to successfully hold meetings and trainings via Zoom due to COVID-19 restrictions, and this modality may have ultimately allowed for higher attendance and therefore richer training and discussion sessions before and during data collection. Despite earlier COVID-19 restrictions, at the time of data collection, the study team was able to visit each store in person to conduct qualitative and quantitative data collection.

The work presented in this special issue is the latest iteration of a series of national partnerships of researchers, supported by the Robert Wood Johnson Foundation/Healthy Eating Research program and the CDC/NOPREN, which are focused on advancing research in community food stores. Previous partnerships have used mixed methods to examine issues in sites throughout the country over the past 15 years, including the study of healthy food availability in small stores [39,40,41,42,43], small store perceptions of the 2009 WIC guidelines [44,45], and the presence of slotting fees and incentives in small food stores [46], all aimed at better understanding the emerging issues where contextualization is of paramount importance.

## 5. Conclusions

In summary, this is the first of a series of papers describing a case study project of seven healthy community food stores. A case study approach was employed in order to provide a deeper, contextually rich understanding of these stores. This first paper laid out the conceptual basis for the study, and described the store recruitment, planning process and methods for data collection and analysis used, in detail. The study seeks to identify factors associated with the success and failure of retailer efforts to offer healthier foods, especially in low-income settings.

## Figures and Tables

**Table 1 ijerph-19-00690-t001:** Summary of the healthy community store case studies: store characteristics.

Location	Date Opened	Financial Model	Store Type	Store Size	Average Healthy Food Availability Index (HFAI)
Baltimore, MD	2018	Non-profit	Grocery store	7000 sq ft	20
Boston, MA	2015	Non-profit	Grocery store	3850 sq ft	20.3
Buffalo, NY	2007	For-profit	Corner store	--	11.6
Chicago, IL	2003	For-profit	Corner Store	3500	12.7
Detroit, MI	1984	For-profit	Supermarket	65,000 sq ft	25
Minneapolis, MN	2015	Co-op	Grocery store	20,000 sq ft	27.5
Washington, DC	2014	For-profit	Market	900 sq ft	19.3

**Table 2 ijerph-19-00690-t002:** Mission, community served, funding of case study stores.

Location	Mission	Community Served	Funding	Current Status
Baltimore, MD	“To provide healthy and affordable food for all members of the community.”	Low–middle-Income; African Americans	A large, international charitable organization	Closed; Feb 2021
Boston, MA	“To provide fresh, tasty, convenient, and nutritious food to communities most in need at prices everyone can afford.”	Low-income; food-insecure; largely immigrant	Funding from 60 funders to date; now 70% covered by revenues	Open
Buffalo, NY	“To offer products that meet the needs of its customers, particularly healthier food options.”	Low-income; African American	Store revenues; state development fund	Open
Chicago, IL	“To create a space for community residents to have access to healthy food items and other necessary things relevant to their everyday use, while providing good customer service.”	Low-income; African American; largely adults under 50	Store revenues	Open
Detroit, MI	“We pride ourselves in offering the best service, quality, and selection of products.”	Low-income; African American	Store revenues	Open
Minneapolis, MN	“To sustain a healthy community that has: equitable economic relationships; positive environmental impacts; and inclusive, socially responsible practices.”	Mix-income; largely African American	Ownership stock; bank loans; co-op owner equity; federal tax credits	Open
Washington, DC	“Developing retail solutions that work in, and for, food desert communities. Through unique partnerships with local growers, producers, and distributors, our experienced retail team is able to offer a full-service grocery selection in a fraction of space.”	Low-income; African American	Store revenue; private foundations; local government	Open

## Data Availability

The data presented in this study are available on request from the corresponding author. The data are not publicly available due to confidentiality considerations.

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
