# Peer review of "Increasing Healthy Food Access for Low-Income Communities: Protocol of the Healthy Community Stores Case Study Project"

_ijerph, 2022, doi:10.3390/ijerph19020690_

Round 1
Reviewer 1 Report
After an exhaustive analysis of the article, I consider it to be of high impact and high level. However, I consider that the results could be significantly improved because they are presented very briefly in this article and perhaps this is one of the sections that should be presented in greater depth and breadth. Therefore, I consider that this section should be expanded in order to accept the present manuscript.
Reviewer 2 Report
This study protocol paper presents the background and methods of the research to understand the experiences of healthy food stores in low-income communities. This paper has nicely framed the research rationale for the study, and aims to answer three research questions. However, while going through the paper repeatedly, I feel that the paper explicitly answers only the second questions, specifically through the Section 2. Materials and Methods. Perhaps the sub-section 'Background of study team' is meant to answer the first research question. I suggest renaming a title of the sub-section and elaborate a bit to explain the origin of the study (or the project). Second paragraph of the sub-section answer the sub-question (What was the justification for conducting a case study analysis?). I believe the content is not well balanced in terms of contents to answer the three research questions.
I found it difficult to find answer for the third question in the submitted paper.
Abstract should also be revised to fit a format of the journal and also to make it coherent with the content.
With these general comments, I have following specific comments.
- Line 193 "... in length from 7-27 pages. Deidentified..." - Why didn't these reports followed the prescribed format as indicated in Appendix 3?
- Line 221 "interesting findings to the group..." - Better to specify at least one or two findings, which authors think are interesting, with some explanation.
- It may be better to start the section 3. Result with the anticipated numbers of papers, papers accomplished and papers in process with clear numbering and labeling.
- Line 232 - Is the first paper already published? If yes how about citing the paper and present some key findings such as the specific strategies for success, challenges, and coping strategies.
Similar is applicable to the second paper. - Line 235-236 needs revision?
- If some papers are already published from this study, I am bit skeptic about the objective of this protocol paper. Since the study (the project) is already under implementation with some outputs accomplished, rather it may be interesting to know what the original research protocol was, how it is been impacted by unprecedented COVID-19 pandemic, and how it is adjusted to cope with the challenges brought by the pandemic.
- Section 4 Discussion & Conclusions, perhaps need an overhauling and concentrate more on the contents of this paper.
- Line 249-250 - Better to refer appendices 1 and 2 in the text as did for Appendix 3.
Reviewer 3 Report
This was an interesting manuscript and seems to be a very worthwhile project to undertake. Please consider the following comments when reviewing the manuscript:
Introduction:
There is a good structure here and the section flows well. Aims are clear. However:
Authors note that only one study so far has investigated this area, might it be useful to explain this out in more detail? It might help with setting a better context to both rationale behind the project and the methodology.
Could a very brief explanation be outlined as to why behavioural interventions (e.g. choice architecture) in existing stores would not / have not worked (particularly within such demographics outlined / focused upon)? This may strengthen the rationale for focusing on healthy food stores / access mission-driven stores specifically.
Method:
Outline of inclusion / exclusion was clear, but line 138 “ the group that demonstrated a more compelling case” leaves ambiguity. How was this judged and by whom?
Were the dropouts replaced by one of the original 18? If so, what was the process for their inclusion?
There is no clear outline of the statistical methods (or qualitative approach for the interviews beyond coding) used for the analysis of data. Could these be added please?
Are there any relevant ethical implications of the protocol that need to be discussed. Although this is more of a feature of a clinical trial study protocol, this would be welcome to the current paper if there were anything noteworthy for the readership.
Suggestion – Could the titling for the individual steps be a little clearly to make it easier to navigate the paper?
Suggestion – if appropriate, consider referring to the supplementary materials in text here. These did help this reviewer get a bit more context to some of the information being explained.
Results:
A concise overview, which may have better context if the above comments are actioned.
Discussion / conclusion:
This section was very repetitive and didn’t provide a sufficient, yet informative, summary of the paper, and in its current form is perhaps redundant. Please consider revising this section, detailing the key points of the rationale, aims and protocol.
Round 2
Reviewer 2 Report
I found most of my comments addressed convincingly.
Best wishes